# Causes of death among people living with metastatic cancer

Kyle Mani[1,6], Daxuan Deng[2], Christine Lin[3,6], Ming Wang [4], Melinda L. Hsu[5] & Nicholas G. Zaorsky [6] ✉

Studying survivorship and causes of death in patients with advanced or metastatic cancer remains an important task. We characterize the causes of death among patients with metastatic cancer, across 13 cancer types and 25 non-cancer causes and predict the risk of death after diagnosis from the diagnosed cancer versus other causes (e.g., stroke, heart disease, etc.). Among 1,030,937 US (1992–2019) metastatic cancer survivors, 82.6% of patients (n = 688,529) died due to the diagnosed cancer, while 17.4% (n = 145,006) died of competing causes. Patients with lung, pancreas, esophagus, and stomach tumors are the most likely to die of their metastatic cancer, while those with prostate and breast cancer have the lowest likelihood. The median survival time among patients living with metastases is 10 months; our Fine and Gray competing risk model predicts 1 year survival with area under the receiver operating characteristic curve of 0.754 (95% CI [0.754, 0.754]). Leading non-cancer deaths are heart disease (32.4%), chronic obstructive and pulmonary disease (7.9%), cerebrovascular disease (6.1%), and infection (4.1%).

As of 2019, cancer is the second leading cause of death in the United States behind cardiovascular disease[1], and the majority of patients who die of cancer die of metastases[2]. However, in recent decades, significant advancements have been made in the fields of cancer prevention, diagnosis, and treatment, both in the United States[3] and Europe[4]. As cancer survivorship rates continue to improve[5,6], it is imperative for key stakeholders in the healthcare industry, including patients, providers, and payers, to identify individuals at the highest risk of mortality, as well as the specific causes of death associated with their cancer diagnosis.

The National Cancer Institute's (NCI's) 2021 Meeting report, authored by various subject matter experts, researchers, clinicians, survivors and advocates, recognized the need to study survivorship in individuals living with metastatic cancers, and specifically referred to this population as an important and often overlooked subpopulation of cancer survivors for which research is limited[7]. Our work aims to fill

the NCI's identified evidence gap of furthering the knowledge of comprehensive survivorship tailored to patients with metastatic disease[7].

The purposes of this work are to (I) characterize the causes of death among patients living with metastatic cancer as a function of disease site, year of diagnosis, and time after diagnosis and (II) predict the risk of death due to diagnosed metastatic cancer versus other causes of death (e.g., stroke, heart disease, etc.) at 1-, 3-, and 5-years after diagnosis. Our overarching goal is to identify patients at highest risk of death from non-cancer causes or original metastatic cancer, as well as those who may benefit from screening for second cancers[8]. Here we report that 688,529 out of 1,030,937 metastatic cancer patients studied died due to the diagnosed cancer, while 145,006 died of competing causes. The median survival time among patients living with metastases is 10 months; our Fine and Gray competing risk model predicts 1 year survival with area under the receiver operating

[1]Albert Einstein School of Medicine, Bronx, NY, USA. [2]Department of Public Health Sciences, Penn State College of Medicine, Hershey, PA, USA. [3]Department of Radiation Oncology, Penn State Cancer Institute, Hershey, PA, USA. [4]Department of Population and Quantitative Health Sciences, School of Medicine, Case Western Reserve University, Cleveland, OH 44106, USA. [5]Division of Hematology and Oncology, University Hospitals Seidman Cancer Center and Case Western Reserve University School of Medicine, Cleveland, OH, USA. [6]Department of Radiation Oncology, University Hospitals Seidman Cancer Center and Case Western Reserve University School of Medicine, Cleveland, OH, USA. ✉e-mail: nicholas.zaorsky@uhhospitals.org

characteristic curve of 0.754 (95% CI: [0.754, 0.754]). Leading non-cancer deaths are heart disease (32.4%), chronic obstructive and pulmonary disease (7.9%), cerebrovascular disease (6.1%), and infection (4.1%). These findings may be used to develop comprehensive guidelines regarding the care of metastatic cancer survivors.

## Results

A total of 1,030,937 patients with newly diagnosed metastatic cancer were abstracted from the SEER database between 1992–2019. 833,535 (80.9%) of these patients died during the follow-up period. The plurality of patients ($n = 688,529$ [82.6%]) died due to the initial diagnosed metastatic cancer and the remaining died of non-cancer cause of deaths ($n = 116,616$ [14.0%]) and the secondary diagnosed cancer ($n = 28,390$ [3.4%]). Table 1 shows baseline co-variates among the study patient population. The high proportion of diagnosed metastatic

deaths remained stable from 1992–2019 (Fig. 1A). Among all patients dying from the initial diagnosed metastatic cancer (Supplementary Fig. 2A), the most had cancers of the lung [32.8%], colon and rectum [7.4%], pancreas [7.4%], ovary [4.4%], and breast [4.0%].

Figure 1B and Supplementary Fig. 3 show the absolute and relative mortality counts from 1992–2019 for the 13 most prevalent metastatic cancers. Patients diagnosed with indolent cancers in recent years are not included in these graphs as they have not yet died from any cause. From 1992 to 2019, there was a slight decrease in relative diagnosed mortality and a slight increase in relative non-cancer mortality among all combined cancers studied (diagnosed cancer deaths: 88.0–86.4% and non-cancer deaths: 9.9–11.4%, Supplementary Fig. 3). There was an increase in relative diagnosed cancer mortality and decrease in non-cancer mortality in patients with newly diagnosed metastatic cancer of the uterine corpus (diagnosed cancer deaths: 75.2–85.3% and

**Table 1 | Baseline co-variates of patients living with metastatic cancer, 1992–2019**

| | Alive (N = 193,126) | Deaths from all causes | | | | Overall (N = 1,030,937) |
|---|---|---|---|---|---|---|
| | | Diagnosed Cancer (N = 688,529) | Non-cancer (N = 111,616) | Subsequent Cancer (N = 28,390) | Unknown (N = 9276) | |
| **Age group** | | | | | | |
| 0–54 | 74,084 (38.4%) | 109,693 (15.9%) | 12,176 (10.9%) | 2662 (9.4%) | 2184 (23.5%) | 200,799 (19.5%) |
| 55–64 | 47,589 (24.6%) | 146,840 (21.3%) | 15,959 (14.3%) | 5038 (17.7%) | 2032 (21.9%) | 217,458 (21.1%) |
| 65–74 | 43,777 (22.7%) | 193,091 (28.0%) | 28,586 (25.6%) | 8793 (31.0%) | 2685 (28.9%) | 276,932 (26.9%) |
| 75–84 | 21,965 (11.4%) | 171,609 (24.9%) | 35,794 (32.1%) | 8790 (31.0%) | 1844 (19.9%) | 240,002 (23.3%) |
| 85+ | 4952 (2.6%) | 66,946 (9.7%) | 19,025 (17.0%) | 3102 (10.9%) | 522 (5.6%) | 94,547 (9.2%) |
| **Sex** | | | | | | |
| Male | 98,051 (50.8%) | 358,696 (52.1%) | 63,664 (57.0%) | 15,404 (54.3%) | 5209 (56.2%) | 541,024 (52.5%) |
| Female | 95,075 (49.2%) | 329,833 (47.9%) | 47,952 (43.0%) | 12,986 (45.7%) | 4067 (43.8%) | 489,913 (47.5%) |
| Race | | | | | | |
| White | 150,372 (77.9%) | 546,256 (79.3%) | 89,663 (80.3%) | 23,473 (82.7%) | 5884 (63.4%) | 815,648 (79.1%) |
| Black | 16,582 (8.6%) | 66,515 (9.7%) | 11,423 (10.2%) | 2558 (9.0%) | 732 (7.9%) | 97,810 (9.5%) |
| Asian or Pacific Islander | 21,946 (11.4%) | 68,426 (9.9%) | 9303 (8.3%) | 2135 (7.5%) | 2488 (26.8%) | 104,298 (10.1%) |
| American Indian/Alaska Native | 1803 (0.9%) | 6452 (0.9%) | 1040 (0.9%) | 215 (0.8%) | 57 (0.6%) | 9567 (0.9%) |
| **Year of diagnosis** | | | | | | |
| 1992–1994 | 3059 (1.6%) | 60,892 (8.8%) | 9478 (8.5%) | 2635 (9.3%) | 817 (8.8%) | 76881 (7.5%) |
| 1995–1999 | 8546 (4.4%) | 112,193 (16.3%) | 17,330 (15.5%) | 4659 (16.4%) | 1328 (14.3%) | 144,056 (14.0%) |
| 2000–2004 | 17,163 (8.9%) | 129,438 (18.8%) | 21,575 (19.3%) | 5189 (18.3%) | 1725 (18.6%) | 175,090 (17.0%) |
| 2005–2009 | 32,199 (16.7%) | 139,949 (20.3%) | 26,441 (23.7%) | 6361 (22.4%) | 1961 (21.1%) | 206,911 (20.1%) |
| 2010–2014 | 52,676 (27.3%) | 142,835 (20.7%) | 23,602 (21.1%) | 6034 (21.3%) | 2009 (21.7%) | 227,156 (22.0%) |
| 2015–2019 | 79,483 (41.2%) | 103,222 (15.0%) | 13,190 (11.8%) | 3512 (12.4%) | 1436 (15.5%) | 200,843 (19.5%) |
| **Primary cancer subsite[a]** | | | | | | |
| Breast | 9376 (4.9%) | 27,778 (4.0%) | 3119 (2.8%) | 667 (2.3%) | 422 (4.5%) | 41,362 (4.0%) |
| Colorectal | 7356 (3.6%) | 51,520 (7.4%) | 4756 (4.3%) | 1302 (4.5%) | 606 (6.5%) | 65,540 (6.4%) |
| Corpus Uteri | 2813 (1.5%) | 8620 (1.3%) | 1148 (1.0%) | 301 (1.1%) | 124 (1.3%) | 13,006 (1.3%) |
| Esophagus | 831 (0.4%) | 11,532 (1.7%) | 767 (0.7%) | 216 (0.8%) | 105 (1.1%) | 13,451 (1.3%) |
| Kidney and Renal Pelvis | 2619 (1.4%) | 17,037 (2.5%) | 1456 (1.3%) | 471 (1.7%) | 206 (2.2%) | 21,789 (2.1%) |
| Liver and Bile Duct | 852 (0.5%) | 11,779 (1.7%) | 1055 (1.0%) | 272 (0.9%) | 231 (2.5%) | 10,653 (1.0%) |
| Lung and Bronchus | 14,833 (7.7%) | 225,999 (32.8%) | 17,827 (16.0%) | 5390 (19.0%) | 2196 (23.7%) | 266,245 (25.8%) |
| Melanoma of the Skin | 1562 (0.8%) | 5099 (0.7%) | 562 (0.5%) | 255 (0.9%) | 51 (0.5%) | 7529 (0.7%) |
| Ovary | 7835 (4.1%) | 30,436 (4.4%) | 2583 (2.3%) | 935 (3.3%) | 366 (3.9%) | 42,155 (4.1%) |
| Pancreas | 2601 (1.3%) | 51,134 (7.4%) | 2374 (2.1%) | 920 (3.2%) | 503 (5.4%) | 57,532 (5.6%) |
| Prostate | 8118 (4.2%) | 22,793 (3.3%) | 6272 (5.6%) | 1060 (3.7%) | 485 (5.2%) | 38,728 (3.8%) |
| Stomach | 2031 (1.1%) | 23,284 (3.4%) | 1428 (1.3%) | 489 (1.7%) | 483 (5.2%) | 27,715 (2.7%) |
| Urinary Bladder | 559 (0.3%) | 6251 (0.9%) | 665 (0.6%) | 300 (1.1%) | 48 (0.5%) | 7823 (0.8%) |

Database "SEER Research Data, 12 Registries, Nov 2021 Sub (1992–2019) was used.
[a]Relative percent is in relation to all 80 primary cancer subtypes abstracted from SEER.

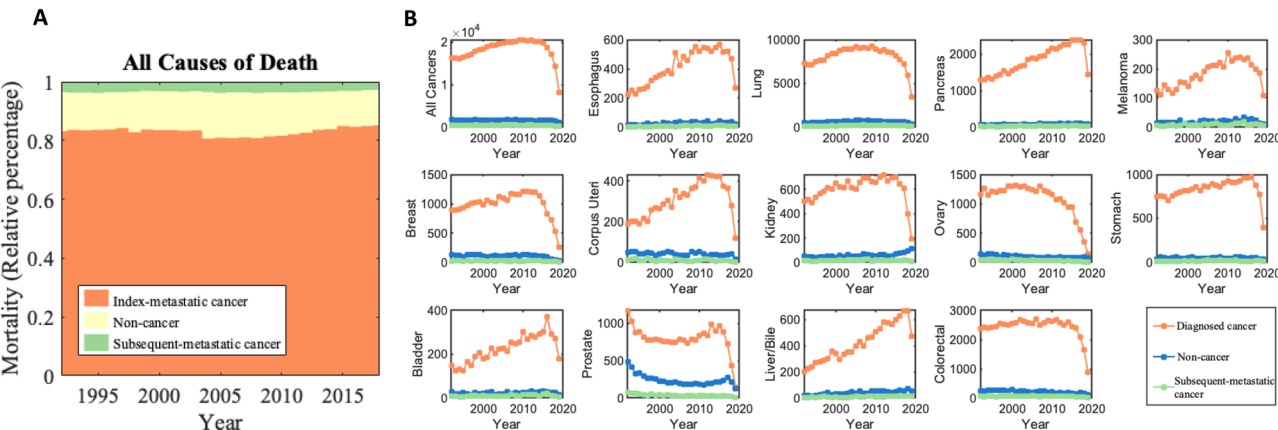

**Fig. 1 | Plots of absolute mortality counts versus year of diagnosis (1992–2019) for various metastatic cancer subtypes. A** Death was stratified due to primary cancer (the cancer originally diagnosed by the patient). **B** Death was stratified due to primary cancer (the cancer originally diagnosed by the patient), secondary cancer, or all other medical causes of death. Patients diagnosed with indolent cancers in recent years are not included in these graphs as they have not yet died from any cause. Source data are provided as a Source Data File.

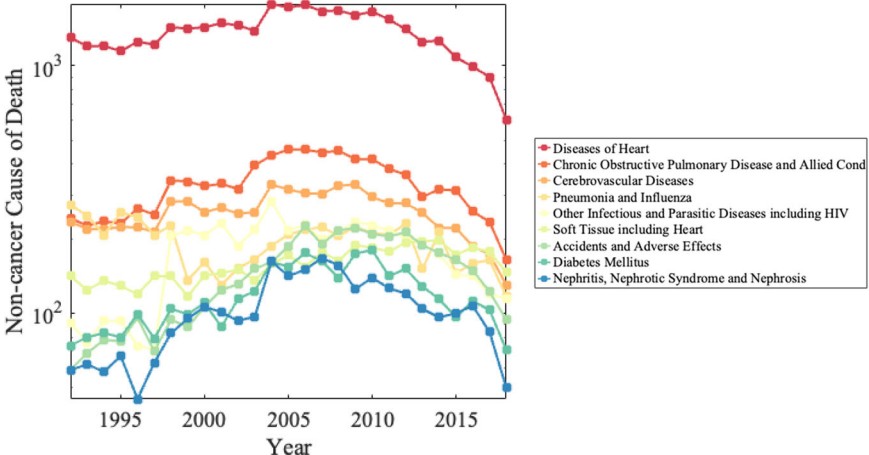

**Fig. 2 | Mortality counts of the top ten contributing non-cancer causes of death among metastatic patients from 1992–2019.** Diseases of Heart (red color), COPD (dark orange color), Cerebrovascular Diseases (light orange color), and Infectious and Parasitic Diseases including HIV (yellow color) were the leading causes of non-cancer death. The decrease in mortality due to non-cancer death in recent years of diagnosis is due to these patients not living long enough to die from a cause of death. Patients diagnosed with indolent cancers in recent years are not included in these graphs as they have not yet died from any cause. Source data are provided as a Source Data File.

non-cancer deaths: 20.1–12.6%) and bladder (diagnosed cancer deaths: 79.1–89.4% and non-cancer deaths: 16.0–8.0%). In prostate cancer patients, diagnosed cancer deaths were the lowest (between 52.7% and 67.5%) while non-cancer deaths were the highest (between 28.1% and 45.6%) (Fig. 1B). Diagnosed cancer deaths (all >80%) have been stable or decreasing in patients with cancers of the esophagus, lung, pancreas, melanocytes, breast, kidney, ovary, stomach, liver/bile duct, and colon and rectal (Supplementary Fig. 3, orange lines).

Currently, the incidence of subsequent-metastatic death is between 1% and 5% in all metastatic cancer subtypes (light green lines among graphs in Supplementary Fig. 3). Supplementary Fig. 4 shows a heatmap of age-adjusted diagnosed metastatic cancer, subsequent-metastatic cancer, and non-cancer death, stratified by 5-year age groups and year of diagnosis. Mortality counts are low for patients <40 years of age while mortality counts are high for patients over the ages of 60 for diagnosed cancer death, 65 for subsequent-metastatic death and 70 for non-cancer death.

Figure 2 illustrates the mortality counts of the top ten non-cancer causes of death among metastatic cancer patients. The leading causes were diseases of the heart (32.4% of non-cancer deaths, 4.4% of all deaths), chronic obstructive pulmonary disease (COPD) (7.9% of non-cancer deaths, 1.1% of all deaths), cerebrovascular disease (6.1% of non-cancer deaths, 0.8% of all deaths), and infectious and parasitic diseases, including HIV (4.1% of non-cancer deaths, 0.6% of all deaths). In 2020, COVID-19 was the second leading non-cancer cause of death (13.0% of non-cancer deaths, 2.6% of all deaths; Supplementary Table 1). When stratifying by SEER's alternative cause of death classification system, ischemic heart disease, deep vein thrombosis and other disorders of the circulatory system, and pulmonary heart disease and embolism contributed to 20.2%, 10.01%, and 0.94% of non-cancer deaths, respectively. Supplementary Fig. 5 shows the relative fatalities from these causes of death by year of diagnosis.

Cerebrovascular diseases are one of the leading causes of death in patients with lung and bronchus, pancreas, melanoma of the skin, corpus uteri, kidney and renal pelvis, prostate, colon and rectum cancers (Supplementary Fig. 6). COPD is prevalent among patients with colon and rectum, kidney/renal pelvis, ovary, urinary bladder, breast, lung and bronchus, and esophagus cancers. Pneumonia and influenza are leading causes of death among prostate and urinary

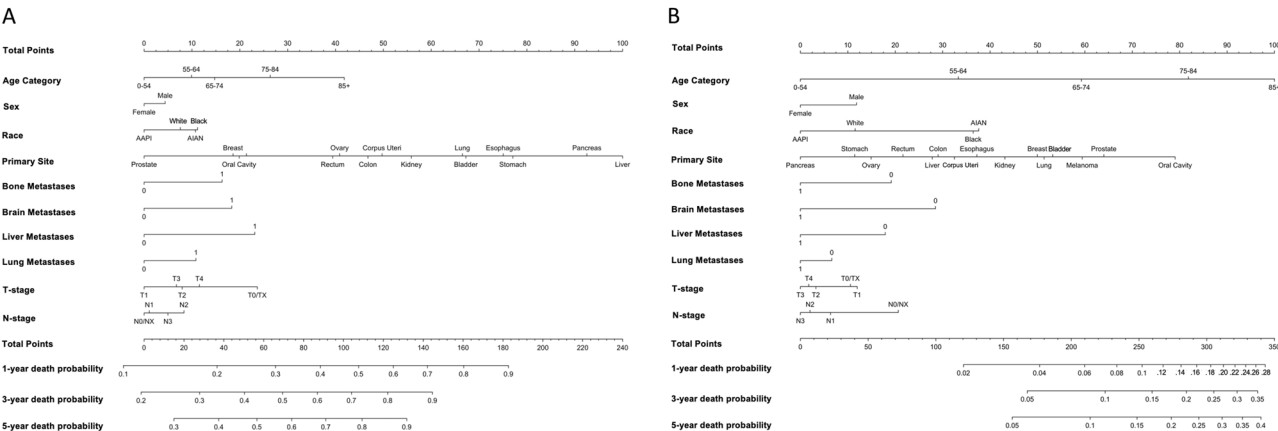

**Fig. 3 | Fine-Gray model nomograms to predict 1, 3, and 5-year survival.**
**A** Nomogram to predict 1, 3, and 5-year cancer-specific mortality using clin-icopathological variables. The model variables include age group (0–54, 55–64, 75–84, and 85+), sex, race, primary cancer site, presence of metastases to the bone, brain, liver, or lung, t-stage, and n-stage. **B** Nomogram to predict 1, 3, and 5-year other-cause mortality using clinicopathological variables. The model variables include age group (0–54, 55–64, 75–84, and 85+), sex, race, primary cancer site, presence of metastases to the bone, brain, liver, or lung, t-stage, and n-stage. Source data are provided as a Source Data File.

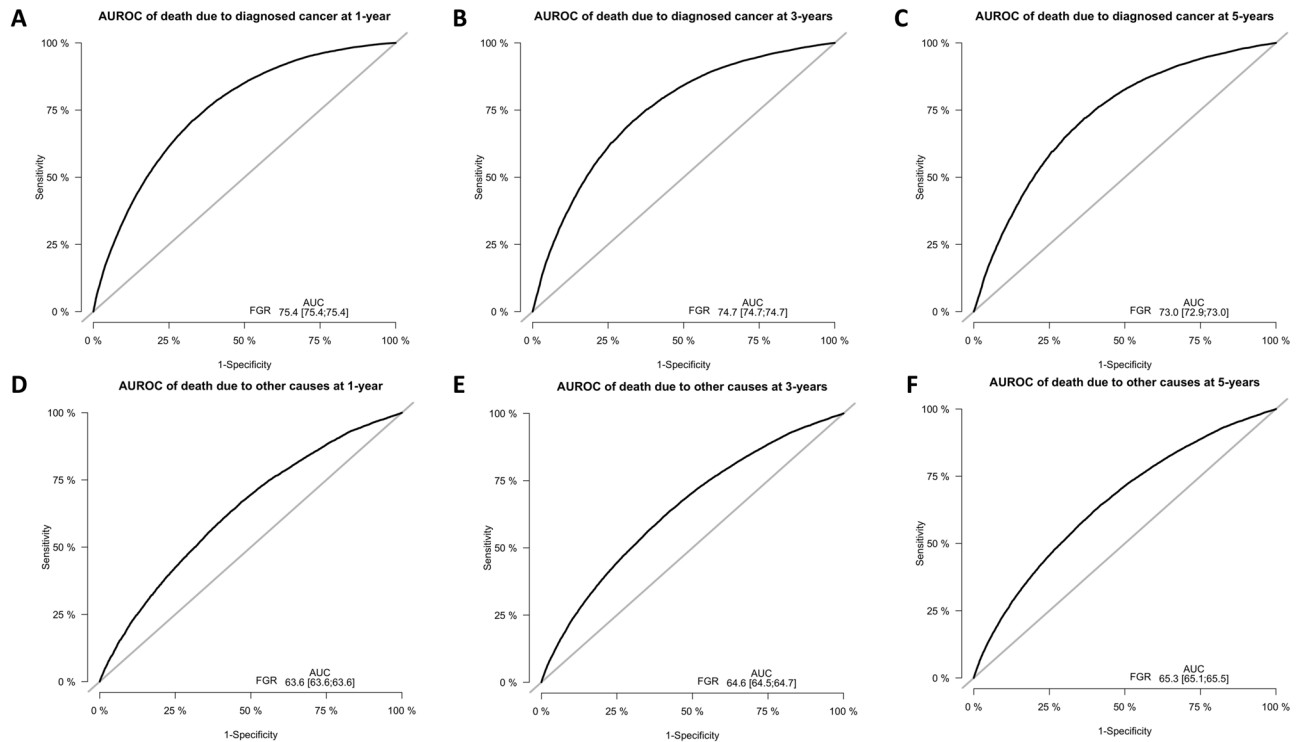

**Fig. 4 | AUROC curves for modeling death due to primary cancer or other cause at 1, 3, and 5-year intervals, respectively.** Sensitivity is plotted on the y-axis and 1-specificity on the x-axis. **A** AUROC of death due to diagnosed cancer at 1-year, **B** AUROC of death due to diagnosed cancer at 3-years, **C** AUROC of death due to diagnosed cancer at 5-years, **D** AUROC of death due to other causes at 1-year, **E** AUROC of death due to other causes at 3-years, **F** AUROC of death due to other causes at 5-years. 95% Confidence Intervals for AUC calculations are included within each plot. The ROC curves for modeling death due to primary cancer demonstrate a good-quality fit. Source data are provided as a Source Data File.

bladder cancer patients. Alzheimer's disease is a leading cause of death in patients with melanoma.

Nomograms were derived and validated based on the competing risks Fine-Gray models to predict 1-, 3-, and 5-year survival probability of death due to diagnosed cancer or other causes in patients with metastatic disease, as shown in Fig. 3. ROC-AUC plots for 1-, 3- and 5-year OS are shown in Fig. 4. Calibration plots for 1-, 3- and 5-year OS are shown in Supplementary Figs. 7 and 8. Parameters include age, sex, race, primary site, presence of metastases to bone, brain, liver, or lung, T-stage, and N-stage. The median survival time among all metastatic cancer patients is 10 months and the AUC of predicting 1-year survival for the initial diagnosed metastatic cancer death from our model is 0.754 (95% CI [0.754, 0.754]). The AUCs of 3-year and 5-year survival for diagnosed cancer death are 0.747 (95% CI [0.747, 0.747]) and 0.730 (95% CI [0.729, 0.730]), respectively. The AUCs of 1-year, 3-year, and 5-year survival for other causes of death are 0.636 (95% CI [0.636, 0.636]), 0.646 (95% CI [0.645, 0.647]) and 0.653 (95% CI [0.651, 0.655]), respectively. The Brier scores of 1-year, 3-year, and 5-year survival for diagnosed cancer death are 0.204 (95% CI [0.203 0.205]), 0.170 (95% CI [0.170, 0.171]) and 0.150 (95% CI [0.150, 0.151]), respectively. The Brier

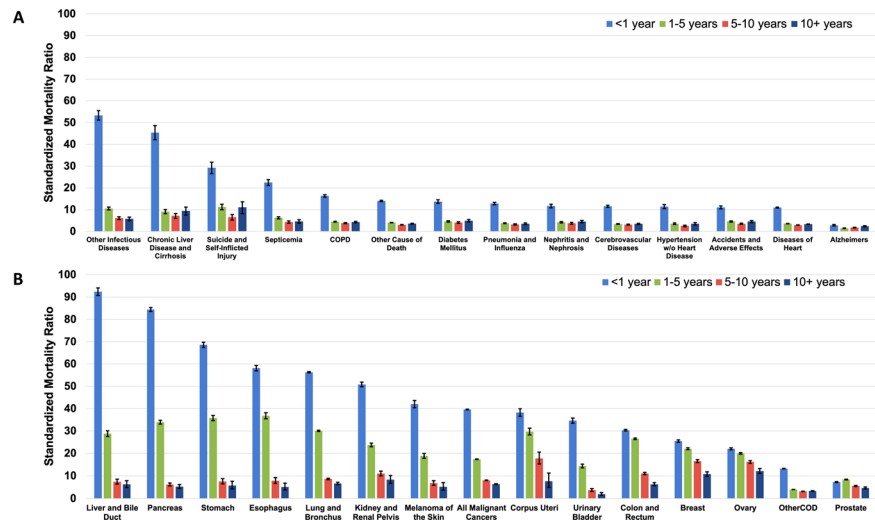

**Fig. 5 | Standardized mortality ratios (SMRs) of non-cancer death and primary-specific cancer subtypes.** The y-axis depicts the SMR with 95% CI, and the x-axis depicts **A** the leading causes of non-cancer death (total person-years at risk = 305,362.9) and **B** primary-specific cancer subtypes (total person-years at risk = 1,631,087.7, stratified by follow-up time. Different time periods after diagnosis (<1 year vs. 1–5 years vs. 5–10 years vs. >10 years) are shown in orange, yellow, green,

and red, respectively. The risk of mortality is highest in the 1st year of diagnosis for all non-cancer deaths and primary-specific cancer subtypes. For most cancers and non-cancer deaths, the SMR subsides with longer follow-up time, but remains greater than the general population. The exact method was used to calculate the 95% CI, and error bars represent the 95% CIs by site. Source data are provided as a Source Data File.

scores of 1-year, 3-year, and 5-year survival for non-cancer death are 0.047 (95% CI [0.046, 0.048]), 0.064 (95% CI [0.063, 0.065]) and 0.072 (95% CI [0.071, 0.073]), respectively.

Standardized mortality ratios (SMRs) for the leading causes of diagnosed cancer and non-cancer death were characterized as a function of time after diagnosis in Fig. 5. Table 2 shows SMRs stratified by noncancer death. SMRs were highest at 1-year follow-up among patients who died of other infectious diseases including HIV (SMR: 53.4, 95% CI: 51.2, 55.6), liver disease and cirrhosis (SMR: 45.5, 95% CI: 42.3, 48.1), suicide and self-inflected injury (SMR: 29.3, 95% CI: 26.8, 31.9), and septicemia (SMR: 22.5, 95% CI: 21.2, 23.9). Among patients dying of diagnosed cancer, SMRs were highest among patients with cancers of the liver/bile duct (SMR: 92.4, 95% CI: 90.7, 94.2), pancreas (SMR: 84.4, 95% CI: 83.6, 85.3), stomach (SMR: 68.6, 95% CI: 67.4, 69.8), and esophagus (SMR: 58.2, 95% CI: 56.9, 59.4).

## Discussion

Historically, metastatic cancer was traditionally considered invariably fatal, with patients inevitably dying of their disease[9]. However, in recent years, more favorable subtypes of metastatic disease have been identified, e.g., STARS IVA-B disease[10] or oligometastases[11]. The NCI acknowledges the need to identify long term survivors and determine causes of death for non-survivors[7]. This information is crucial in tailoring patient care to prevent death from specific causes. Our study broadly characterizes and provides predictive insight into the cause of death among patients diagnosed with metastatic cancer. We report that (I) 79.2% of patients with metastatic disease died from their diagnosed cancer, while 17.1% died of competing causes, and this trend has remained stable since the 1990s; and (II) the median survival time for all cancer patients with metastases is 10 months, and our predictive model achieved a high AUC (0.754 (95% CI [0.754, 0.754]) for 1-year survival.

Non-cancer deaths among patients who live with metastatic cancer may largely be divided into two groups: (I) chronic comorbid conditions or (II) acute, iatrogenic, or treatment-induced infections. Metastatic cancer patients are over 10 times more likely than the general population to die from heart (32.4% of non-cancer deaths) or cerebrovascular disease (6.1% of non-cancer deaths) in the first year after diagnosis. These findings may be due to high risk of

cardiotoxicity from aggressive treatment among those with co-existing CVDs[12–15]. The American Heart Association (AHA) and National Comprehensive Cancer Network recently recognized the impact of cancer treatment on cardiovascular health, with a focus on cardio-oncology[16,17] and emphasized the need for multi-modal cardiac rehabilitation programs for cancer survivors[18]. However, metastatic cancer patients are not specifically discussed, despite the large heart disease burden.

COPD was the second most prevalent cause of death among in lung, breast, and colorectal cancer patients (7.9% of all non-cancer deaths); however, lung and breast associations have overlooked the mortality risk due to COPD. The American Lung Association[19] (ALA) and the International Association of the Study of Lung Cancer (IASLC)[20] provide guidelines for respiratory disease prevention for the general population and those with comorbidities, but not specifically for patients with metastases. The Breast Cancer Research Foundation's recent investment in metastatic breast cancer research[21] aims to improve patients' quality of life and understand risk factors for metastasis, but does not address COPD, which is the second leading non-cancer cause of death among this patient population. As the fields of cardio-oncology and pulmonary-oncology develop, we recommend that the AHA and ALA provide comprehensive guidelines for the care of cancer patients living with metastases.

Notably, patients with metastatic cancer also have 14.6 times higher of suicide than that of the public, which may be due to depression[22,23] and feelings of hopelessness[24] associated with a poor-prognosis cancer diagnosis. Although screening for distress is widely recommended for all cancer patients by major medical professional organizations, the implementation of these recommendations has been low[25–29]. Additionally, these distress screenings are only validated in patients receiving cancer care; there are no validated tools for cancer survivors who may not be on active treatment. Strategies to prevent suicide may be targeted at metastatic cancer patients with lung, prostate, and colorectal cancer (Supplementary Fig. 9) and should improve access to support groups and regular distress screening in the first year after diagnosis. Notably, this study only included intentional self-harm as a cause of suicide, and physician-assisted suicide was not included. Therefore, it possible that the true

**Table 2 | Standardized mortality ratios of noncancer death among metastatic cancer patients**

| Cause of death | Non-cancer death counts[a] | Survival (Months)[a] | Person years at risk[b] | SMR (95% CI)[b] |
|---|---|---|---|---|
| Accidents and adverse effects | 3849 | 24 | 12,557.2 | 5.4 (5.2–5.6) |
| Alzheimer's (ICD-9 and 10 only) | 2536 | 65 | 13,162.0 | 1.9 (1.8–2.0) |
| Aortic aneurysm and dissection | 526 | 16 | 1441.9 | 5.8 (5.2–6.4) |
| Atherosclerosis | 683 | 10 | 1476.8 | 5.0 (4.5–5.4) |
| Cerebrovascular diseases | 6870 | 19 | 19,875.2 | 4.8 (4.6–4.9) |
| Certain conditions originating in perinatal period | 21 | 1 | 28.6 | 62.3 (36.9–98.5) |
| Chronic liver disease and cirrhosis | 1649 | 8 | 3959.8 | 15.4 (14.6–16.2) |
| Chronic obstructive pulmonary disease | 8949 | 14 | 22,757.5 | 6.6 (6.4–6.7) |
| Complications of pregnancy and childbirth | 101 | 7 | 136.1 | 50.6 (40.8–62.2) |
| Congenital anomalies | 255 | 8 | 517.8 | 13.8 (11.9–15.8) |
| Diabetes mellitus | 3190 | 21 | 9962.6 | 6.2 (5.9–6.4) |
| Diseases of heart | 36696 | 18 | 100,714.2 | 4.7 (4.7–4.8) |
| Homicide and legal intervention | 83 | 24 | 291.0 | 30.2 (23.7–37.9) |
| Hypertension without heart disease | 1683 | 22 | 5346.8 | 4.5 (4.3–4.8) |
| Nephritis, nephrotic syndrome and nephrosis | 2751 | 24 | 8602.4 | 5.5 (5.2–5.7) |
| Other cause of death | 20804 | 18 | 61,803.4 | 5.1 (5.1–5.2) |
| Other diseases of arteries, arterioles, capillaries | 554 | 18 | 1621.4 | 5.2 (4.7–5.7) |
| Other infectious and parasitic diseases and HIV | 4633 | 7 | 8644.3 | 16.7 (16.2–17.2) |
| Pneumonia and influenza | 5261 | 15 | 13,552.6 | 5.4 (5.2–5.5) |
| Septicemia | 2608 | 11 | 5825.7 | 8.7 (8.3–9.1) |
| Soft tissue including heart | 4310 | 9 | 5491.8 | 38.5 (37.2–39.7) |
| Stomach and duodenal ulcers | 456 | 8 | 898.0 | 8.5 (7.7–9.4) |
| Suicide and self-inflicted injury | 1233 | 10 | 2699.5 | 14.6 (13.7–15.6) |
| Symptoms, signs and ill-defined conditions | 1847 | 11 | 3875.0 | 6.4 (6.1–6.8) |
| Tuberculosis | 66 | 11 | 121.4 | 10.0 (7.5–13.1) |

[a]Database "SEER Research Data, 12 Registries, Nov 2021 Sub (1992–2019) was used for death counts and survival months.
[b]Database "SEER Research Data, 12 Registries (excl AK), Nov 2021 Sub (1992–2019) for SMRs" was used.

overall incidence of suicide among patients living with cancer is higher than reported in this study.

We report that COVID-19 was the second leading non-cancer cause of death among all patients living with metastatic cancer in 2020, behind ischemic heart disease, and notably higher than other well-studied causes, such as stroke[30] and suicide[31]. The year 2020 presented unique circumstances regarding causes of death among cancer patients, primarily due to (I) reduced overall cancer reporting[32–35] and (II) the initial deployment of COVID-19 vaccines not occurring until December of 2020[36]. The findings of our study can be interpreted as capturing the highest risk period for fatal COVID-19 infection in individuals living with cancer prior to widespread vaccine adoption. Our findings are still highly relevant in the US population, as a sizeable minority of US residents do not plan to get a COVID-19 vaccine due to geopolitical factors, such as misinformation, distrust in healthcare providers, personal ideologies, and cost, as reported by the Lancet Commission on Vaccine Refusal, Acceptance, and Demand[37]. Furthermore, patients with cancer have been found to have an increased risk of COVID-19 infection[38] and a more severe disease course[39], warranting further research in this uniquely vulnerable population.

There are several strengths to this study. The 2022 NCI's meeting report highlighted the importance of ongoing epidemiological and surveillance research among survivorship for individuals living with advanced or metastatic cancers[7]. While previous analyses have investigated the causes of death in patients with specific metastatic cancers (e.g. prostate[40,41] and liver[42]), or among cancer patients who have died from a particular cause of death (e.g. suicide[31], cardiovascular disease[43], fatal heart disease[44]), the present study characterizes causes of death among 13 individual metastatic cancers, from 25 major types of non-cancer death, as a function of calendar year and follow-up time.

This study also aimed to fulfill the NCI's goal of developing and testing models of comprehensive survivorship among patients living with metastatic cancer[7]. We leveraged Fine-Gray competing risk models and deployed an easy-to-use clinical risk tool that can predict 1-year survival with AUC of 0.754 (95% CI: 0.754, 0.754) and is available online for prediction (http://tinyurl.com/met-mortality). In comparison, our recent analysis demonstrated that American Joint committee on Cancer (AJCC) staging system can predict 1-year survival with AUC of 0.641 (95% CI: 0.638, 0.646) in NCDB patients[10]. While further external validation of our model is necessary, our web-based calculator may be used by clinicians to estimate a patient's risk of mortality due to metastatic cancer. The 10 variables required are readily available at the hospital or out-patient setting. If the patient's risk is low, the clinician may opt for a decrease in intense follow ups (eg, with visits, imaging) and preferential referral to competing risk clinics (e.g., onco-cardiology, onco-pulmonology, and onco-nephrology)[43–45], whereas high-risk estimations may support adherence to the NCCN's palliative care guidelines[46].

Risk-prediction tools are typically designed for populations rather than individuals. While these tools can provide practitioners with an estimated likelihood of complications in a patient, it is important to recognize that each patient is unique and influenced by factors that are not always captured or evaluated in clinical practice. The average risks of patients under the care of an individual clinician may be influenced by local practice referral patterns or individual practice styles. As SEER patients are treated by numerous clinicians across the country, there is a higher likelihood of accounting for this variability and developing a robust risk-prediction tool applicable in most clinical settings.

This study also has important limitations. There is a potential for misclassification of deaths caused by the diagnosed cancer, which can occur because of attributing a local or regional relapse to a new cancer or metastasis to a nearby organ. Additionally, some non-cancer-related deaths may be attributed to treatment, which were attempted to be identified through patient age and time after diagnosis. The SEER database was established in 1973 and gradually accrued over time, which may result in an increased risk of death among patients in the database due to a smaller denominator of all cancer patients. This is also influenced by the fact that patients diagnosed in earlier years have longer follow-up times and are at greater risk for death, while patients diagnosed more recently have shorter follow-up times and lower likelihood of death from any cause.

Our study provides predictive insight into the cause of death among patients newly diagnosed with metastatic cancer. Approximately 80% of patients living with metastatic cancer will die of their diagnosed cancer, while 20% will die of competing causes (heart disease, COPD, stroke, subsequent cancer deaths in >50% of these patients). This has remained consistent for 30 years. We created a simple nomogram with AUC of 0.75 and deployed an online calculator to predict which patients will die of their diagnosed cancer. We present a framework for predicting the most likely causes of death, which may serve as a valuable tool for clinicians in determining the most effective interventions for individual patients.

## Methods

The present study utilizes a three-part analytical strategy, as detailed in Supplementary Fig. 1. Patient data for those diagnosed with metastatic cancer between 1992 and 2019 were extracted from the National Cancer Institute's Surveillance, Epidemiology, and End Results (SEER) program (SEER 18 database)[47]. The methods and limitations for each component of the analysis are outlined in the Supplementary Methods. To evaluate the impact of COVID-19 on mortality patterns, a detailed analysis of mortality counts of the 26 non-cancer causes of death among patients with active follow-up in 2020 is presented in Supplementary Table 1. These 26 causes include the 25 non-cancer causes of death in the primary analysis and COVID-19. The SEER program is a network of population-based incident tumor registries from various regions in the United States, representing 28% of the country's population, and includes information on incidence, survival, and treatment (such as radiation therapy, surgery, and chemotherapy)[47]. However, the SEER registry does not include information on comorbidities, performance status, surgical pathology, margin status, doses, or systemic agents. Patients diagnosed only through autopsy or death certificate were excluded from the study. We comply with all relevant ethical regulations. These data are freely available and thus the study was exempt from institutional review board review. There are no participants in the study, and thus no consent form.

The strategy for objective I is shown in Supplementary Fig. 1. Mortality was classified as being a result of the "diagnosed metastatic death" (the cancer originally diagnosed in the patient), "subsequent-metastatic death" (a secondary primary cancer), and "non-cancer death" (death from any medical cause not coded as cancer) using certificate data. For objective II, the analysis of death was conducted in accordance with established methods for suicide[6,48] and included a total of 25 non-cancer causes of death. The top 13 causes of death were reported, utilizing standardized mortality ratios (SMRs) as a measure. SMRs provide the relative risk of death for patients with cancer compared to the general US population[48,49]. The data was characterized with SMRs adjusted for age, race (as reported by SEER in line with the National Center of Health Statistics), and sex to the US population over the same time period. The 95% confidence intervals (CIs) of the SMRs were calculated using SEER*Stat 8.4.1 and Microsoft Excel 16.0.1 (Microsoft, Redmond, WA)[49–51]. All other data analysis was conducted

using MATLAB R2022b (MathWorks, Inc., Natick, MA), and R Studio 3.3.0+ (R Studio Inc., Boston, MA).

Mortality codes in SEER are assigned from death certifications, filed by the doctor caring for the patient at the time of death. For the purposes of this study, patient mortality was based on coding in the International Classification of Diseases for Oncology 3rd (ICD-O-3) edition codes (Supplementary Table 2). Non-cancer causes of death were classified into 25 unique diseases as follows: (1) Accidents and Adverse Effects; (2) Alzheimer's; (3) Aortic Aneurysm and Dissection; (4) Atherosclerosis; (5) Cerebrovascular Diseases; (6) Certain Conditions Originating in Perinatal Period; (7) Chronic Liver Disease and Cirrhosis; (8) Chronic Obstructive Pulmonary Disease; (9) Complications of Pregnancy and Childbirth; (10) Congenital Anomalies; (11) Diabetes Mellitus; (12) Disease of Heart; (13) Homicide and Legal Intervention; (14) Hypertension without Heart Disease; (15) Nephritis, Nephrotic Syndrome and Nephrosis; (16) Other Cause of Death; (17) Other Diseases of Arteries, Arterioles, Capillaries; (18) Other Infectious and Parasitic Diseases and HIV; (19) Pneumonia and Influenza; (20) Septicemia; (21) Soft Tissue including Heart; (22) Stomach and Duodenal Ulcers; (23) Symptoms, Signs and Ill-Defined Conditions; (24) Suicide and Self-Inflicted Injury; (25) Tuberculosis.

Additionally, we provide an alternative cause of death classification system using SEER's latest ICD-O-3 2023+ cause of death recodes and the corresponding mortality counts (1992–2019) in Supplementary Table 3. The main purpose of including this alternative classification is to improve granularity in reporting cause of death for broad groupings such as *Diseases of the Heart* (ICD-10: I00-I09, I11, I13, I20-I51), which is now redistributed as follows: (1) Ischemic Heart Disease (ICD-10: I20-I25), (2) Pulmonary Heart Disease and Disease of Pulmonary Circulation (ICD-10: I20-I25), (3) Other forms of Heart Disease (ICD-10: I00-I09, I30-I15, I80-I99). The corresponding clinical descriptions of these alternative ICD-O-3 codes can be found in Supplementary Table 4.

To generate nomograms Fine-Gray sub distribution hazard regression models were fit based on clinical and demographic data, such as age, sex, race, primary cancer type, and the presence of metastasis to the bone, brain, liver, and/or lung, similar to a recent pan-cancer analysis of metastatic phenotypes[10]. The model development cohort consisted of all patients diagnosed with metastatic cancer from 2010 to 2019 ($n = 466,404$). To assess the predictive accuracy of the models, we employed a 5-fold cross-validation resampling technique[52]. As per current practice in survival literature, the competing risk models were validated using the time-dependent Uno's concordance (C) index[53] and Brier score[54] to quantify discrimination and calibration, respectively, at 1–3-and 5-year intervals of the uncensored population event times. A weighted dataset for competing risks was created[55,56], and the final Fine-Gray competing risk models were then trained on this entire dataset to develop nomograms[57]. For a detailed mathematical description, please refer to the original paper[58]. Weighted scores were assigned to clinical prediction variables based on the model's variable coefficients (Supplementary Table 5). The prediction tool incorporates these variables from the final model, assigning a specific point total to each variable. The results were then used to create nomograms to predict the corresponding risk of death in patients living with metastatic cancer for the initial diagnosed metastatic cancer and non-cancer causes of death at 1, 3, and 5 years after diagnosis. The Fine and Gray models were then deployed as a clinical application using RShiny and are available online for external prediction (http://tinyurl.com/met-mortality).

### Reporting summary

Further information on research design is available in the Nature Portfolio Reporting Summary linked to this article.

## Data availability

The relevant session information used to abstract data and the user-submitted request and abbreviated data set (from SEER) for SMRs are provided in the Supplementary File[47]. The individual patient-level SEER data are protected and are not available due to data privacy laws[47]. However, the SEER patient data is publicly available under restricted access; access can be obtained via application form in compliance with relevant National Cancer Institute research use data agreements in the following repository: (https://seer.cancer.gov/data/access.html)[47]. The remaining data are available within the article and Supplementary Information file. Source data are provided with this paper.

## Code availability

The Fine and Gray survival calculator generated in this study can be accessed for external prediction and have been deposited under accession code: http://tinyurl.com/met-mortality. All other code used to perform statistical analyses or to train and validate the Fine and Gray models and the source code for the prediction calculator can be accessed at the following publicly available repository at ZENODO: https://doi.org/10.5281/zenodo.10428551[59]

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

## Author contributions

All authors had full access to all of the data in the study and take responsibility for the integrity of the data and the accuracy of the data analysis. Study concept and design: KM, NZ. Acquisition, analysis, and interpretation of data: KM, DD, MW, KM. Drafting of the manuscript: KM. Critical revision of the manuscript for intellectual content: KM, DD, CL, MW, MH, NZ. Statistical analysis: KM, DD, MW. Obtained funding: N/A. Administrative, technical, or material support: N.Z. Study supervision: NZ.

## Competing interests

The authors declare no competing interests.
