## [Peer Review File · Nature Communications]

REVIEWER COMMENTS

Reviewer #1 (Remarks to the Author): Expert in cancer epidemiology and statistics, and cancer mortality

-While the authors say there was an improvement in cancer treatment in recent years, the relative diagnosed cancer mortality increased. Any explanations for that?

-How the COVID-19 pandemic may affect the causes of death among cancer patients should be analyzed and discussed.

-The authors did not discuss how the predictive model built in this study can be used in clinical settings.

-Rather than focus on the discussion on suicide, the authors should discuss the novel findings of this study.

-References 36-39 did not match those in the text.

-Sensitivity analyses should be conducted to exclude patients diagnosed with cancers in the recent five years

-The training and validation sets should be randomly divided rather than by period. This may have caused bias over time.

Reviewer #3 (Remarks to the Author): Expert in oncology, epidemiology, health and therapy outcomes

This is an interesting, and very thorough analysis of the SEER database, which aims to characterize and predict the causes of death among patients with metastatic cancer across 13 cancer types and 25 non-cancer causes. The study analyzed data from over a million US metastatic cancer survivors from 1992 to 2019. The authors find that ~83% of patients died due from their cancer, while ~17% died of competing causes. Patients with lung, pancreas, esophagus, and stomach tumors were likely to die of their metastatic cancer, while those with prostate and breast cancer were least likely. The median survival

time among patients living with metastatic cancer was 10 months the authors also developed model that predicted 1-year survival with good accuracy.

This is an appropriate use of the SEER database and to my knowledge; this comprehensive of a report on metastatic cancer has not been published. The manuscript is well-written. The methods are appropriate and explained well. Limitations acknowledged include misclassification. A few comments:

Pulmonary embolism I suspect is a fairly common cause of death among cancer patients due to hypercoagulability but seems to be missing from causes of death? All I see is cerebrovascular disease. If DVT/PE is not accounted for in this database, that is major limitation/omission.

Does suicide include physician-assisted dying? If this is not identifiable, that is another limitation and may inflate the suicide statistics.

Regression models that inform the nomogram are missing. Typically, the underlying multivariable models are presented, including coefficients and measures of the model's overall fit and predictive accuracy. This is critical in understanding the relative contribution of each variable in predicting survival probabilities, and for assessing the validity and robustness of the model.

REVIEWER COMMENTS

Reviewer #1 (Remarks to the Author): Expert in cancer epidemiology, statistics, and mortality

-While the authors say there was an improvement in cancer treatment in recent years, the relative diagnosed cancer mortality increased. Any explanations for that?

AUTHOR RESPONSE: Thank you for your helpful comments. In the results (**Lines 40-48**), we stated that the relative diagnosed cancer mortality increased only for cancers of the uterine corpus and bladder. However, **Supplementary Figure 3** shows that cancer mortality remained stable or decreased in the other 11 studied cancer subtypes. Additionally, the relative diagnosed cancer mortality for all combined cancers slightly decreased from 1992 to 2019 (**Supplementary Figure 3**). These findings support literature that cancer treatment has improved, albeit modestly, in patients living with metastatic cancer.

SUPPLEMENTARY FIGURE 3

We have edited the text in the results to further clarify that relative diagnosed mortality for all cancers has slightly decreased (**Lines 37-48**):

“From 1992 to 2019, there was a slight decrease in relative diagnosed mortality and a slight increase in relative non-cancer mortality among all combined cancers studied (diagnosed cancer deaths: 88.0% to 86.4% and non-cancer deaths: 9.9% to 11.4%, **Supplemental Figure 3**). There was an increase in relative diagnosed cancer mortality and decrease in non-cancer mortality in patients with newly diagnosed metastatic cancer of the uterine corpus (diagnosed cancer deaths: 75.2% to 85.3% and non-cancer deaths: 20.1% to 12.6%) and bladder (diagnosed cancer deaths: 79.1% to 89.4% and non-cancer deaths: 16.0% to 8.0%). In prostate cancer patients, diagnosed cancer deaths were the lowest (between 52.7% and 67.5%) while non-cancer deaths were the highest (between 28.1% and 45.6%) (**Figure 1B**). Diagnosed cancer deaths (all >80%) have been stable or decreasing in patients with cancers of the esophagus, lung, pancreas, melanocytes, breast, kidney, ovary, stomach, liver/bile duct, and colon and rectal (**Supplemental Figure 3**, orange lines).”

Additionally, our analysis includes patients who are diagnosed each year, adds patients who were alive from prior years, and excludes patients who have died of various causes (**Supplementary Figure 1**, right panel). Patients diagnosed in recent years have not lived long enough to die of either their diagnosed cancers or competing causes, which explains any rapid changes at the tail end of the time period (<5 years). We report a similar finding among all patients living with cancer, in our previous paper: Zaorsky NG, Churilla TM, Egleston BL, et al. Causes of death among cancer patients. *Ann Oncol.* 2017;28(2):400-407. doi:10.1093/annonc/mdw604.

-How the COVID-19 pandemic may affect the causes of death among cancer patients should be analyzed and discussed.

AUTHOR RESPONSE: Thank you for your helpful comments. To supplement the current study, we have conducted an additional exploratory analysis and included a **Supplemental Table 1** that characterizes non-cancer causes of death during 2020 including COVID-19. Taken together, **Table 2** and **Supplemental Table 1** may further our understanding of the causes of death in patients living with metastatic cancer prior to the COVID-19 pandemic (2000-2019) and after the onset of the COVID-19 pandemic (2020). Our exploratory analysis showed that in 2020, COVID-19 (2.6% of overall deaths and 13.0% of all non-cancer causes of death) ranked as the 2nd leading non-cancer cause of death behind heart disease.

We now cite **Supplemental Table 1** in the methods (**Lines 210-214**):

“To evaluate the impact of COVID-19 on mortality patterns, a detailed analysis of mortality counts of the 26 non-cancer causes of death among patients with active follow-

SUPPLEMENTARY FIGURE 1

Objective: Characterizing the causes of death among patients living with metastatic cancer

- Total deaths: 845,648
 - Diagnosed-cancer: 693,279
 - Subsequent-cancer: 28,566
 - Non-cancer cause: 114,445
 - Heart disease: 37,209

Objective: Characterizing the top 25 non-cancer causes of death among patients living with metastatic cancer

- Non-cancer cause: 114,445
 - Accidents and Adverse Effects
 - Alzheimer’s (ICD-9 and 10 only)
 - Aortic Aneurysm and Dissection
 - Atherosclerosis
 - Cerebrovascular Diseases
 - Conditions in Perinatal Period
 - Chronic Liver Disease and Cirrhosis
 - COPD
 - Complications of Pregnancy
 - Congenital Anomalies
 - Diabetes Mellitus
 - Diseases of Heart
 - Homicide and Legal Intervention
 - Hypertension without Heart Disease
 - Nephritis and Nephrotic Syndrome
 - Other Cause of Death
 - Diseases of Arteries, Arterioles, Capillaries
 - Infectious, Parasitic Diseases, HIV
 - Pneumonia and Influenza
 - Septicemia
 - Soft Tissue including Heart
 - Stomach and Duodenal Ulcers
 - Suicide and Self-Inflicted Injury
 - Symptoms and Ill-Defined Conditions
 - Tuberculosis

up in 2020 is presented in **Supplemental Table 1**. These 26 causes include the 25 non-cancer causes of death in the primary analysis and COVID-19.”

We now report pertinent findings in the results (**Lines 61-62**):

“In 2020, COVID-19 was the second leading non-cancer cause of death (13.0% of non-cancer deaths, 2.6% of all deaths; **Supplemental Table 1**).”

We now elaborate on these findings in the discussion (**Lines 145-157**):

“We report that COVID-19 was the second leading non-cancer cause of death among all patients living with metastatic cancer in 2020, behind ischemic heart disease, and notably higher than other well-studied causes, such as stroke²⁹ and suicide.³⁰ The year 2020 presented unique circumstances regarding causes of death among cancer patients, primarily due to (I) reduced overall cancer reporting³¹⁻³⁴ and (II) the initial deployment of COVID-19 vaccines not occurring until December of 2020.³⁵ The findings of our study can be interpreted as capturing the highest risk period for fatal COVID-19 infection in individuals living with cancer prior to widespread vaccine adoption. Our findings are still highly relevant in the US population, as a sizeable minority of US residents do not plan to get a COVID-19 vaccine due to geopolitical factors, such as misinformation, distrust in healthcare providers, personal ideologies, and cost, as reported by the Lancet Commission on Vaccine Refusal, Acceptance, and Demand.³⁶ Furthermore, patients with cancer have been found to have an increased risk of COVID-19 infection³⁷ and a more severe disease course,³⁸ warranting further research in this uniquely vulnerable population.”

Please also note that in the current work, we aimed to characterize causes of death (diagnosed cancer vs. subsequent cancer vs. competing causes) among patients living with metastatic cancer. The dataset we used at the time of analysis spanned from 1992 to 2019, before the COVID-19 variable was made available in SEER on April 19th, 2023. Currently, 2020 is the only year with COVID-19 mortality data available in a time period that precedes the widespread availability of vaccines. Thus, a direct comparison, or inclusion of this data into the primary analysis, is not possible using SEER. Our group has just submitted a comprehensive analysis and evaluation of death due to COVID-19 vs. cancer for the year 2020; this is outside of the scope of the current manuscript, and we have inserted all relevant data regarding death from COVID among patients with metastatic disease.

-The authors did not discuss how the predictive model built in this study can be used in clinical settings.

AUTHOR RESPONSE: Thank you for your helpful comments. In the discussion, we have clarified how a secondary aim of our study was to fulfill the NCI’s goal of developing and testing models of comprehensive survivorship among metastatic cancer patients. We now also include text in the discussion (**Lines 166-186**) to describe how the predictive model can be used in clinical settings:

“This study also aimed to fulfill the NCI’s goal of developing and testing models of comprehensive survivorship among patients living with metastatic cancer.⁷ We

leveraged Fine-Gray competing risk models and deployed an easy-to-use clinical risk tool that can predict 1-year survival with AUC of 0.754 (95% CI: 0.754, 0.754) and is available online for prediction (<http://tinyurl.com/met-mortality>). In comparison, our recent analysis demonstrated that American Joint committee on Cancer (AJCC) staging system can predict 1-year survival with AUC of 0.641 (95% CI: 0.638, 0.646) in NCDB patients.¹⁰ While further external validation of our model is necessary, our web-based calculator may be used by clinicians to estimate a patient's risk of mortality due to metastatic cancer. The 10 variables required are readily available at the hospital or out-patient setting. If the patient's risk is low, the clinician may opt for a decrease in intense follow ups (eg, with visits, imaging) and preferential referral to competing risk clinics (e.g., onco-cardiology, onco-pulmonology, and onco-nephrology),⁴²⁻⁴⁴ whereas high-risk estimations may support adherence to the NCCN's palliative care guidelines.⁴⁵

Risk-prediction tools are typically designed for populations rather than individuals. While these tools can provide practitioners with an estimated likelihood of complications in a patient, it is important to recognize that each patient is unique and influenced by factors that are not always captured or evaluated in clinical practice. The average risks of patients under the care of an individual clinician may be influenced by local practice referral patterns or individual practice styles. As SEER patients are treated by numerous clinicians across the country, there is a higher likelihood of accounting for this variability and developing a robust risk-prediction tool applicable in most clinical settings.”

-Rather than focus on the discussion on suicide, the authors should discuss the novel findings of this study.

AUTHOR RESPONSE: Thank you for your comments. We have revised the discussion (**Lines 166-186**) to focus on the paragraph/point above. We re-iterate how our work addresses the two National Cancer Institute goals regarding further research in survivorship among metastatic cancer patients, including further discussion focused on the novelty of our predictive model and how it can be deployed in the clinical setting.

We now also discuss our findings (**Lines 145-157**, above) regarding mortality attributed to COVID-19 among metastatic cancer patients in the year 2020. Notably, we report that COVID-19 was the 2nd leading non-cancer cause of death among patients living with metastatic cancer in 2020, after heart disease.

-References 36-39 did not match those in the text.

AUTHOR RESPONSE: Thank you for your comments. We have corrected the references, so that they match accordingly. We have also carefully looked at all other citations in the text, including those added in our revisions, to ensure that they match those in the reference section.

Below, we have included the corrected text and provided the updated citations (now references 30, 42, and 43) in question:

“While previous analyses have investigated the causes of death in patients with specific metastatic cancers (e.g. prostate^{39,40} and liver⁴¹), or among cancer patients who have died from a particular cause of death (e.g. suicide,³⁰ cardiovascular disease,⁴² fatal heart disease⁴³), the present study is unique in that it characterizes causes of death among 13 individual metastatic cancers, from 25 major types of non-cancer death, as a function of calendar year and follow-up time.”

30. Zaorsky, N. G. et al. *Suicide among cancer patients. Nature Communications* 2019 10:1 10, 1–7 (2019).

42. Sturgeon, K. M. et al. *A population-based study of cardiovascular disease mortality risk in US cancer patients. Eur Heart J* 40, 3889–3897 (2019).

43. Stoltzfus, K. C. et al. *Fatal heart disease among cancer patients. Nature Communications* 2020 11:1 11, 1–8 (2020).

-The training and validation sets should be randomly divided rather than by period. This may have caused bias over time.

AUTHOR RESPONSE: Thank you for your comments. We have retrained our model using a resampling method of 5-fold cross-validation (five iterations of randomly partitioned data into 80:20 train/test splits) to calculate average AUCs. This widely implemented machine-learning technique allows us to assess the generalizability of our model and is superior to the one-time splitting method, which may raise concerns related to the potential bias introduced by model evaluation on a single randomly partitioned train-test split.

After training on randomly partitioned data, our calculated AUC values demonstrated slight improvements at 1-, 3- and 5-years follow-up compared to our previously reported AUCs trained on temporally split datasets. We have updated **Figures 3 and 4** and the AUC values throughout the manuscript, accordingly.

In the methods (**Lines 259-274**), we have written the following:

“The model development cohort consisted of all patients diagnosed with metastatic cancer from 2010 to 2019 (n = 466,404). To assess the predictive accuracy of the models, we employed a 5-fold cross-validation resampling technique.⁵⁰ As per current practice in survival literature, the competing risk models were validated using the time-dependent Uno's concordance (C) index⁵¹ and Brier score⁵² to quantify discrimination and calibration, respectively, at 1-3- and 5-year intervals of the uncensored population event times. A weighted dataset for competing risks was created,^{53,54} and the final Fine-Gray competing risk models were then trained on this entire dataset to develop novel nomograms.⁵⁵ For a detailed mathematical description, please refer to the original paper.⁵⁶ Weighted scores were assigned to clinical prediction variables based on the model's variable coefficients (**Supplemental Table 5**). The prediction tool incorporates these variables from the final model, assigning a specific point total to each variable. The results were then used to create nomograms to predict the corresponding risk of death in patients living with metastatic cancer for the initial diagnosed metastatic cancer and non-cancer causes of death at 1, 3, and 5 years after diagnosis. The Fine and Gray models were then deployed as a clinical

application using RShiny and are available online for external prediction (<http://tinyurl.com/met-mortality>).”

-Sensitivity analyses should be conducted to exclude patients diagnosed with cancers in the recent five years

AUTHOR RESPONSE: Thank you for your comments. We have now implemented several measures to ensure robustness and perform additional sensitivity analyses on our model. As an exploratory analysis, we calculated AUCs for a test cohort that excluded patients diagnosed with metastatic cancers within the past 5 years. For predicting death due to diagnosed cancer, we report comparable AUCs of 0.750 (95% CI: 0.750, 0.750), 0.739 (95% CI: 0.738, 0.739), and 0.715 (95% CI: 0.715, 0.715), at 1-, 3-, and 5-years, respectively. In the main text, however, we only report the results of the 5-fold cross validation, which is a more reliable and widely accepted estimate of a model's performance.

We have also produced calibration curves (**Supplemental Figures 7 and 8**) as another form of sensitivity analysis. The calibration curves provide a graphical representation of the agreement between the predicted and observed outcomes across the range of predicted probabilities and show that our model is well-calibrated. Last, as per current practice in survival literature, we calculate the time-dependent Brier score (**Supplemental Figures 7 and 8**) to quantify calibration in addition to the C-index that we previously reported for discrimination at the dataset-specific 1-, 3-and 5-year intervals of the uncensored population event times. By calculating Brier scores, we can further quantitatively demonstrate that our model is well calibrated.

SUPPLEMENTARY FIGURE 7

SUPPLEMENTARY FIGURE 8

Reviewer #3 (Remarks to the Author): Expert in oncology, epidemiology, health and therapy outcomes

This is an interesting, and very thorough analysis of the SEER database, which aims to characterize and predict the causes of death among patients with metastatic cancer across 13 cancer types and 25 non-cancer causes. The study analyzed data from over a million US metastatic cancer survivors from 1992 to 2019. The authors find that ~83% of patients died due from their cancer, while ~17% died of competing causes. Patients with lung, pancreas, esophagus, and stomach tumors were likely to die of their metastatic cancer, while those with prostate and breast cancer were least likely. The median survival time among patients living with metastatic cancer was 10 months the authors also developed model that predicted 1-year survival with good accuracy.

This is an appropriate use of the SEER database and to my knowledge; this comprehensive of a report on metastatic cancer has not been published. The manuscript is well-written. The methods are appropriate and explained well. Limitations acknowledged include misclassification. A few comments:

AUTHOR RESPONSE: Thank you for your kind comments and helpful feedback, which has greatly improved our work.

Pulmonary embolism I suspect is a fairly common cause of death among cancer patients due to hypercoagulability but seems to be missing from causes of death? All I see is cerebrovascular disease. If DVT/PE is not accounted for in this database, that is major limitation/omission.

AUTHOR RESPONSE: Thank you for your comments. To improve the clarity of our reporting, we have added a **Supplemental Table 2** which includes the ICD-10 codes and clinical descriptions for the NCI's 25 non-cancer causes of death that were included in this analysis. **Supplemental Table 2** shows that the classification of DVT/PE is grouped in *Diseases of the Heart* and is

accounted for in our analysis. DVT/PE is a significant contributor to *Diseases of the Heart* and is in part why this category is the leading non-cancer cause of death among this patient cohort.

On April 19th, of 2023, SEER added an alternative cause of death variable. For completeness, we have added **Supplemental Tables 3 and 4**, which show total mortality counts from 2000-2019, stratified by the new alternative cause of death recode, and the corresponding ICD-10 codes + clinical descriptions, respectively. It is worth noting that the alternative recode variable is currently not used by the NCI to officially report cause of death, nor is it used in the NCI's yearly cancer statistics review. However, the juxtaposition of **Supplemental Tables 2-4** with **Table 2** (which uses the NCI's standard COD recode, as does the rest of our analysis) can provide additional granularity of the causes of death included in this work. For example, the new recode separates ischemic heart disease from DVT/PE. **Supplemental Table 3** shows that PE contributed 0.94% and DVT + other disease of the circulatory system contributed 10.01% of all non-cancer causes of death, respectively.

We have edited the methods (**Lines 233-235**) to mention **Supplemental Table 2**:

“For the purposes of this study, patient mortality was based on coding in the International Classification of Diseases for Oncology 3rd (ICD-O-3) edition codes (**Supplemental Table 2**).

We have edited the methods (**Lines 247-255**) to mention **Supplemental Tables 3 and 4** and incorporate the new cause of death classification system:

“Additionally, we provide an alternative cause of death classification system using SEER's latest ICD-O-3 2023+ cause of death recodes and the corresponding mortality counts (1992-2019) in **Supplemental Table 3**. The main purpose of including this alternative classification is to improve granularity in reporting cause of death for broad groupings such as *Diseases of the Heart* (ICD-10: I00-I09, I11, I13, I20-I51), which is now redistributed as follows: (1) ischemic heart disease (ICD-10: I20-I25), (2) Pulmonary Heart Disease and Disease of Pulmonary Circulation (ICD-10: 126-128), and (3) Other forms of heart disease (ICD-10: I00-I09, I30-I51, I80-I99). The corresponding clinical descriptions of these alternative ICD-O-3 codes can be found in **Supplemental Table 4**.”

We have edited the results (**Lines 62-65**) to highlight DVT/PE as important causes of death:

“When stratifying by SEER's alternative cause of death classification system, ischemic heart disease, deep vein thrombosis and other disorders of the circulatory system, and pulmonary heart disease and embolism contributed to 20.2%, 10.01%, and 0.94% of non-cancer deaths, respectively.”

Does suicide include physician-assisted dying? If this is not identifiable, that is another limitation and may inflate the suicide statistics.

AUTHOR RESPONSE: Thank you for your comments. To improve clarity and maintain full transparency we have included a **Supplemental Table 1** that has ICD-10 codes and brief

descriptions of the 25 non-cancer causes of death included in this analysis. The ICD-10 codes for suicide are U03, X60-X84, Y870 and these correspond to intentional self-harm. Since physician-assisted suicide is not included, our results do not inflate suicide reporting. Alternatively, it is likely that the true incidence of suicide is actually higher than reported in this study.

Ultimately, the granularity necessitated for a meaningful sub-analysis would be better served for another paper. This work is also described in detail in “*Suicide among cancer patients*” in *Nature Comms* (Jan. 2019).

We have updated the discussion section (**Lines 141-144**) as follows:

“Notably, this study only included intentional self-harm as a cause of suicide, and physician-assisted suicide was not included. Therefore, it possible that the true overall incidence of suicide among patients living with cancer is higher than reported in this study.”

Regression models that inform the nomogram are missing. Typically, the underlying multivariable models are presented, including coefficients and measures of the model's overall fit and predictive accuracy. This is critical in understanding the relative contribution of each variable in predicting survival probabilities, and for assessing the validity and robustness of the model.

AUTHOR RESPONSE: Thank you for your comments. We now include a **Supplemental Table 5** that includes the adjusted sub-distribution hazard ratio coefficients and *p*-values for each variable included in our Fine and Gray competing risk models. The juxtaposition of **Supplemental Table 5** and the nomograms in **Figure 3** provide both numeric and visual methods of quantifying the Fine-Gray competing risk model. To provide an alternative metric of the model’s overall validity, we calculated the Brier score, which quantifies calibration and discrimination, in addition to our previously reported AUCs. To further assess the robustness of the model, a 5-fold cross-validation resampling method was performed to report the average AUC and Brier score for 1, 3 and 5-year intervals, and calibration curves were plotted for our model (**Supplemental Figures 7 and 8**).

We believe that we have addressed all the concerns of the reviewers. If there are any additional concerns, please do not hesitate to contact us.

REVIEWERS' COMMENTS

Reviewer #1 (Remarks to the Author):

The paper has been improved.

Reviewer #3 (Remarks to the Author):

The authors have written a very thoughtful and thorough response. All of my concerns have been addressed.

REVIEWER COMMENTS

Reviewer #1 (Remarks to the Author): Expert in cancer epidemiology, statistics, and mortality

The paper has been improved.

AUTHOR RESPONSE: Thank you for your thorough feedback, which has greatly improved our work.

Reviewer #3 (Remarks to the Author): Expert in oncology, epidemiology, health and therapy outcomes

The authors have written a very thoughtful and thorough response. All of my concerns have been addressed.

AUTHOR RESPONSE: Thank you for your thorough feedback, which has greatly improved our work.

We believe that we have addressed all the concerns of the reviewers. If there are any additional concerns, please do not hesitate to contact us.